# Integrated Management of Persistent Atrial Fibrillation

**DOI:** 10.3390/biomedicines13010091

**Published:** 2025-01-02

**Authors:** Xindi Yue, Ling Zhou, Chunxia Zhao

**Affiliations:** 1Division of Cardiology, Tongji Hospital, Tongji Medical College, Huazhong University of Science and Technology, Wuhan 430074, China; 15083125100@163.com; 2Hubei Key Laboratory of Genetics and Molecular Mechanisms of Cardiological Disorders, Wuhan 430074, China; zhouling_tjh@163.com

**Keywords:** persistent atrial fibrillation, catheter ablation, risk factor

## Abstract

The global incidence of atrial fibrillation is on the rise. Atrial fibrillation, a complex disease, heightens the likelihood of heart failure, stroke, and mortality, necessitating careful attention. Controlling heart rate and rhythm, addressing risk factors, and preventing strokes are fundamental in treating atrial fibrillation. Catheter ablation stands out as the primary approach for atrial fibrillation rhythm control. Nevertheless, the limited success rates pose a significant challenge to catheter ablation, particularly for persistent atrial fibrillation. Various adjunctive ablation techniques are currently under investigation to enhance the effectiveness of catheter ablation. This review provides an overview of the current state of the art and the latest optimized treatments for persistent atrial fibrillation in the areas of rhythm control, heart rate control, and risk factor management.

## 1. Introduction

Atrial fibrillation (AF) is the most common sustained cardiac arrhythmia [1], affecting 33 million people worldwide. AF incidence is expected to rise by over 60% by 2050 [2]. The mechanisms of AF are currently controversial, with significant differences among patients [3]. Paroxysmal AF terminates spontaneously, typically within a few hours to a few days. Rapid excitement from pulmonary vein usually initiates it. Persistent atrial fibrillation (PeAF) is more closely related to atrial fibrosis. Trigger foci often initiate AF, while atrial remodeling promotes the persistence of AF, as seen in Figure 1. Pulmonary vein isolation (PVI) catheter ablation has emerged as an effective AF treatment, especially for individuals with paroxysmal AF. However, the success rate for PeAF remains relatively low. Several ablation techniques and strategies for PeAF are currently under exploration. This review aims to offer an overview of the present status of rhythm control, heart rate control, and risk factor optimization in the management of atrial fibrillation.

## 2. Risk Factor Management in Persistent Atrial Fibrillation

Prompt identification and intervention of these risk factors can not only slow the advancement of atrial arrhythmic substrates and lower the occurrence of persistent atrial fibrillation but also enhance the effectiveness of rhythm control in persistent atrial fibrillation. Risk factors for atrial fibrillation are shown in Figure 2.

Obesity: Obesity—defined as a body mass index (BMI) of ≥30 kg/m^2^—is strongly associated with the onset and advancement of AF [5]. Adipose tissue induces left atrial fibrosis and electrical remodeling through mechanisms such as inflammation and oxidative stress, contributing to the onset of AF. Managing weight loss was proven to reverse the gradual progression of AF. The REVERSE-AF study indicated that among patients with persistent AF who shed 10% of their body weight, 88% transitioned to paroxysmal AF or no AF [6]. Conversely, elevated BMI also accelerates the shift from paroxysmal AF to persistent AF. The experimental results of Mahajan et al may explain the mechanistic reasons for this shift. In a model of 30 obese sheep, weight loss (30%) was associated with reverse remodeling of cardiac structure and electrophysiology. This was evidenced by decreased left atrial pressure, reduced left atrial fibrosis and inflammation, and increased effective atrial sufficiency and conduction velocity in the weight-loss group [7]. Sustained and stable weight loss promotes maintenance of sinus rhythm and reduces AF burden in patients with atrial fibrillation. Weight fluctuations greater than 5% rather doubles the risk of arrhythmia recurrence [8]. The reason may be that weight fluctuations increase the risk of hypertension, diabetes, and other cardiometabolic traits [9,10,11].

Dietary modification is an important part of managing obesity. The Mediterranean diet primarily consists of plant-based foods, with animal products limited to fish and a small amount of dairy. The PREDIMED (Prevención con Dieta Mediterránea) trial found a Mediterranean diet enriched with extra virgin olive oil reduced the risk of AF by 38% [12], highlighting its role in primary prevention. Furthermore, adherence to this diet significantly lowered epicardial adipose tissue in AF patients, a factor linked to poor prognosis in persistent AF. Reducing epicardial adipose tissue may be one mechanism through which the Mediterranean diet protects against AF [13].

Alcohol Consumption: Chronic heavy alcohol consumption can lead to changes in atrial remodeling and vagal activity, potentially triggering atrial fibrillation [14,15]. There is a dose-response association between alcohol intake and AF risk, with an overall linear trend [16]. Moderate to heavy alcohol consumption raises the risk of progressing to persistent atrial fibrillation [17]. There is an ongoing debate about the impact of low alcohol consumption on AF. The HUNT (Nord-Trøndelag Health) study found that consuming < 1 drink daily only slightly raised AF risk while consuming > 2 drinks/day significantly increased the risk [18]. However, a 2021 observational study indicated alcohol consumption in patients with AF heightened the risk of ischemic stroke, independent of the quantity consumed [19]. Alcohol cessation, as part of a comprehensive atrial fibrillation management strategy, lowers the risk of arrhythmia recurrence in AF patients who consume alcohol regularly and improves the quality of life [20], it should be further promoted.

Smoking: Smoking is a significant reversible risk factor for atrial fibrillation. This prospective population-based study from Rotterdam found that both current and former smokers have an elevated risk of developing the condition [21]. A meta-analysis by Aune et al. revealed a dose-dependent relationship between cigarette consumption and the risk of AF. Current smokers exhibited the highest risk for developing AF, followed by former smokers, while ever smokers showed the lowest association [22]. An observational study suggests that smoking raises the risk of atrial fibrillation recurrence in patients with persistent AF undergoing pulmonary vein isolation (PVI) [23]. Possible reasons may be because smoking promotes the development of non-pulmonary vein triggers and also has a chronic effect of promoting atrial fibrosis [24,25,26]. Smoking also heightens the risk of adverse outcomes in patients with atrial fibrillation. In a cohort study of 7329 AF patients on anticoagulation therapy, smoking doubled the stroke risk [27]. Quitting smoking is a matter of concern. Alongside raising awareness, considering nicotine replacement medication may be beneficial if needed.

Hypertension: Hypertension stands as the most common risk factor for AF [28]. Recent Mendelian randomized studies have affirmed the causal link between elevated blood pressure and AF risk [29]. The 2016 Framingham Heart Study reveals patients with persistently elevated systolic blood pressure face double the Risk of Atrial Fibrillation over 15 years [30]. Early treatment of hypertension can be used for the primary prevention of atrial fibrillation. In a post hoc analysis of the SPRINT trial, intensive antihypertensive treatment decreased the risk of new-onset AF by 26% [31]. Controlling systolic blood pressure in patients with isolated systolic hypertension resulted in a 17% reduction in new-onset atrial fibrillation, according to the LIFE randomized controlled trial [32]. Uncontrolled hypertension is an independent risk factor for recurrence after atrial fibrillation ablation, while controlled hypertension does not affect AF ablation outcomes similarly to patients without hypertension [33]. The optimal therapeutic target for managing AF hypertension is uncertain, but evidence indicates that overly aggressive blood pressure reductions may be detrimental. The 2017 SMAC-AF (Substrate Modification with Aggressive Blood Pressure Control) trial randomized patients into two groups: one with aggressive blood pressure control (<120/80 mm Hg) and another with standard control (<140/90 mm Hg). Aggressive management did not lower the recurrence of atrial arrhythmias after catheter ablation compared to standard therapy. Instead, it increased the risk of adverse events, including symptomatic hypotension [34]. Renal sympathetic denervation is a novel therapy used to treat refractory hypertension. Recent studies suggested renal sympathetic denervation can inhibit atrial remodeling, shorten inactivity duration, slow atrial conduction velocity, and reduce neurohumoral activation [35,36,37,38]. In the ERADICATE trial, renal sympathetic denervation catheter ablation significantly lowers recurrence rates of paroxysmal atrial fibrillation after ablation without increasing procedural complications [39]. The role of this therapy in patients with persistent atrial fibrillation requires further validation in clinical studies.

Obstructive sleep apnea (OSA): AF and OSA are common comorbidities [40]. OSA increases the risk of atrial fibrillation through hypoxia, oxidative stress, autonomic and humoral regulation disturbances, and atrial stretching [41,42,43]. Chrishan J et al. found that continuous positive airway pressure (CPAP) therapy reversed atrial remodeling in patients with AF and OSA. Patients were randomized in a 1:1 ratio to receive either CPAP or conventional treatment. At follow-up, those in the CPAP group exhibited higher atrial voltages, faster conduction velocities, and fewer complex points [44]. Several observational clinical studies have shown that treating OSA properly could reduce the recurrence rate and burden of AF [45,46,47]. However, when researchers attempted to test this hypothesis in randomized trials, CPAP therapy did not improve rhythm outcomes in patients with AF [48,49]. The association between CPAP and improved AF outcomes in patients with OSA remains controversial. Furthermore, poor compliance with CPAP therapy limits the use of this therapy. Future research is needed to determine optimal thresholds for OSA severity to identify patients who would benefit most from CPAP therapy.

Autonomic activation plays a role in the pathogenesis of AF in OSA. In acute OSA, increased sympathetic and vagal activity triggers AF episodes, and sympathetic activation in chronic OSA patients results in cardiac remodeling associated with AF [41,50]. The autonomic nervous system may be a potential therapeutic target for patients with OSA combined with AF. A series of preclinical studies have shown that autonomic modulation techniques such as cardioselective β-blockers [51,52], renal denervation [53,54], low-level vagal nerve stimulation [55], ganglionated plexi ablation [50,56,57] have a protective effect on OSA-related AF.

Diabetes: Diabetes raises the risk of AF [58,59], which in turn further elevates the risk of stroke and death in patients with type 2 diabetes. Given the increasing prevalence of type 2 diabetes globally, interventions against these risk factors are particularly important. Glucagon-like peptide-1 receptor agonists (GLP-1 RAs) are mainly used for the treatment of type 2 diabetes mellitus. Increasing evidence supports the cardioprotective effects of GLP-1 RA and their role in improving myocardial metabolism [60]. According to Loryn J et al.’s animal model, GLP-1 treatment in type 2 diabetic mice could prevent atrial conduction impairment, electrical remodeling, and fibrosis, thereby reducing the risk of atrial fibrillation [61]. A 2021 network meta-analysis evaluating the effect of glucose-lowering drugs on the risk of AF/atrial flutter in diabetic patients showed that GLP-1 RA significantly reduced the risk of AF/atrial flutter compared to other glucose-lowering agents [62]. However, some findings suggest minimal benefit of GLP-1 RA use in reducing the risk of atrial fibrillation [63,64], possibly due to the varied mechanisms involved in AF development [60]. Clinical consensus on the ability of GLP-1 RAs to reduce AF risk has not yet been reached. Satti et al. investigated the impact of GLP-1 RA on AF recurrence and found that using GLP-1 RA before ablation did not decrease the risk of AF recurrence post-ablation [65]. This contrasts with Abu-Qaoud et al. who reported that sodium-glucose cotransporter-2 inhibitors significantly reduced AF recurrence risk before ablation [66]. Various classes of glucose-lowering drugs affect the risk of atrial fibrillation differently, necessitating further research to explore these effects more comprehensively.

Physical activity: Moderate-intensity continuous training significantly reduces the incidence of atrial fibrillation and improves the prognosis of patients with atrial fibrillation [67]. Furthermore, RCT shows that a short-term exercise training program improved symptoms, exercise capacity, and quality of life in patients with persistent atrial fibrillation compared to a control group [68]. This conclusion was further extended by Kato et al. in 2019: moderate exercise training in patients with persistent AF post-ablation improves exercise capacity, reduces inflammatory markers, and does not increase the risk of AF recurrence [69].

## 3. Rate Control

The ventricular rate significantly impacts hemodynamic outcomes and symptoms in atrial fibrillation patients [70]. For those with persistent AF unlikely to maintain sinus rhythm, heart rate control may be the preferred treatment. Medications such as beta-blockers, digoxin, verapamil, or diltiazem are commonly prescribed to slow the ventricular rate by inhibiting conduction at the AV node [71]. Some physicians aim for stricter heart rate control, targeting a mean resting heart rate of <80 beats/min. However, research shows no variance in adverse events between strict and lenient heart rate control groups in permanent atrial fibrillation patients (strict: target heart rate < 80 beats/min at rest and <110 beats/min with moderate exercise; lenient: target heart rate < 110 beats/min) [72]. Therefore, a slightly higher resting heart rate is acceptable in the absence of significant symptoms and normal cardiac function.

In patients with persistent AF who do not respond to medication and are unsuitable for catheter ablation, the approach combining atrioventricular node ablation with pacemaker implantation should be considered [71]. Yet, complications from chronic pacemaker stimulation have been reported to significantly impact patients’ quality of life. Right ventricular pacing may cause non-physiologic ventricular excitation and impair ventricular function, leading to worsening of heart failure [73]. In contrast, biventricular pacing is less prone to ventricular asynchronous contractions [74,75]. However, in patients with narrow QRS duration (QRS < 130 ms), biventricular pacing can prolong ventricular excitation time and may cause asynchronous ventricular contractions [76]. His-Purkinje conduction system pacing (HPSP) could leverage the heart’s intrinsic conduction system to achieve physiological ventricular activation [77,78,79]. Clinical studies demonstrate a high success rate for HPSP paired with AV node ablation in heart failure patients with persistent atrial fibrillation, leading to significant improvement in left ventricular function and a reduction in adverse events [80,81,82]. HPSP combined with atrioventricular node ablation presents a valuable treatment option for patients with persistent AF and heart failure who do not respond to heart rate and rhythm control therapies.

## 4. Rhythm Control

Recently, the EAST-AFNET 4 trial randomized patients with early-stage atrial fibrillation (history < 1 year) to an early rhythm control group (antiarrhythmic drugs or ablation) or a conventional treatment group [83]. Results showed that patients in the early rhythm control group had a lower risk of a major composite outcome than those in the conventional treatment group. While the primary goal of rhythm control is to alleviate symptoms and enhance quality of life, it can also improve prognosis in certain patient groups. However, it is important to note that the EAST-AFNET 4 trial mainly included patients with paroxysmal AF, and there are currently no early rhythm control trials focused on persistent AF. The potential benefits of rhythm control for this population require further investigation. AAD and catheter ablation are the main methods for managing AF rhythms.

### 4.1. Drug Rhythm Control

Antiarrhythmic drugs (AADs) are typically recommended as first-line treatment for AF. The limited efficacy and relatively common side effects of AADs have restricted their use [84]. To minimize the side effects of AADs, the choice of AADs should be tailored based on factors such as patient comorbidities, age, gender, drug interactions, and other considerations [71]. Drugs like flutamide, propafenone, and d-sotalol are linked to a higher risk of mortality in patients with structural heart disease, and dronedarone increases mortality risk in those with heart failure. However, these drugs can be used in patients without structural heart disease [85]. Amiodarone is considered the most effective antiarrhythmic drug [1] and can be prescribed to patients with compromised left heart function. Yet, due to its potential adverse effects (hepatotoxicity, pulmonary fibrosis, etc.), long-term use should be avoided when possible [86].

The effectiveness of catheter ablation in persistent AF is unsatisfactory. Persistent AF is a long-standing condition often linked to atrial fibrosis, with PV and non-PV trigger foci that develop over time [87]. Relevant studies have shown that the success rate of persistent AF ablation is approximately 70% after one year [88,89,90]. The efficacy of PVI drops to 40–50% after five years [91,92,93,94]. Thus, exploring novel catheter ablation strategies for persistent AF is essential.

### 4.2. Empiric Approaches

(1).Posterior wall isolation (PWI)

The left atrial posterior wall, sharing embryonic origin with pulmonary veins [95,96], exhibits an arrhythmogenic electrophysiological structure. Targeting PWI for isolation is common for adjunctive ablation. This approach offers potential benefits like autonomic ganglia, rotor, and fibrotic region modification [97]. Examining the effectiveness and safety of PVI + PWI adjunctive ablation is essential. Encouraging results from some small randomized controlled trials confirm the additional benefit of adjunctive PWI in improving procedure outcomes in persistent atrial fibrillation [96,98]. However, the 2023 CAPLA trial, comparing PVI-only and PVI + PWI groups in persistent AF, found no significant difference in atrial fibrillation recurrence rates 12 months post-procedure [99]. Similarly, in the 2019 POBI-AF study, patients with persistent AF were randomly assigned to PVI alone or PVI with posterior wall Box Isolation. After a mean follow-up of 16.2 ± 8.8 months, there was no significant difference in the rate of AF recurrence between the two groups [100]. The results of two large randomized controlled trials do not back the use of PWI during the initial ablation of persistent AF. Due to anatomical proximity, PWI poses risks of damage to the esophagus. Given the risk of esophageal injury, the above study may not have strictly adhered to the protocol when isolating the posterior wall. Furthermore, some researchers suggest that individual patient factors (e.g., atrial fibrillation severity) may influence PWI efficacy. Specific subgroups (abnormal left atrial structure, low posterior wall voltage, and long-term persistent AF) benefit more from PWI + PVI [101]. Conversely, PWI offers limited advantages for persistent AF patients with normal left atrial structure. Randomized studies of PWI efficacy in persistent AF are summarized in Table 1. More research is necessary to ascertain PWI’s benefits for particular patient subgroups (abnormal left atrial structure, low posterior wall voltage, and long-term persistent AF) [101]. A 2021 meta-analysis discovered a higher AT/AFL recurrence rate with RF-based PWI compared to CB-based procedures. The uniform lesion formation by CB over point-to-point RF ablation could explain this discrepancy [102]. However, it is unclear whether the lower recurrence rate in cryoablation is related to posterior wall reconnection. Pulsed electrical field is an emerging form of ablative energy. In 2020, Reddy et al. utilized pulsed field energy to ablate the posterior wall of the left atrium. An invasive remapping assessment 2–3 months later demonstrated 100% durable isolation of the left atrial posterior wall [103]. The above findings indicate that ablation energy may have a role to play in isolating the posterior wall effectively.

(2).Vein of Marshall ablation(3).The abundant innervation, myocardial connections, and arrhythmic foci in the vein of Marshall (VOM) make it an ideal target for AF ablation. Retrograde balloon cannulation from the coronary sinus to the VOM and infusion of ethanol enables rapid ablation of adjacent myocardium and innervation. Research indicates that combining pulmonary vein isolation with VOM ethanol infusion enhances ablation success in persistent atrial fibrillation. In the VENUS trial, persistent AF patients were randomly assigned to PVI alone or PVI plus vein of Marshall ethanol infusion. There are higher success rates of sinus rhythm maintenance in the PVI plus Marshall ethanol infusion group [108]. The subsequent MARSHALL-Plan trial further explored the effectiveness of a combined ablation strategy (Marshall ablation, PVI, and linear ablation) in patients with persistent atrial fibrillation and patients in the combined-strategy ablation group had up to 79% AF/AT recurrence-free rate at 12 months [109]. Of note, the location of the Marshall vein in the mitral isthmus makes it strongly associated with perimitral atrial tachycardia. Achieving a bi-directional blockade of the mitral isthmus is currently the main challenge in catheter ablation of persistent atrial fibrillation. Based on data from long-term follow-up outcomes in atrial fibrillation, researchers found that VOM ethanol infusion facilitates mitral isthmus ablation [110]. Furthermore, a recent study confirmed that VOM ethanol infusion can reduce the risk of acute reconnection after mitral isthmus block [111]. Therefore, VOM ethanol infusion can be regarded as a valuable complement to mitral isthmus ablation.Ganglionated plexi ablation:

The autonomic nervous system influences the onset and progression of AF. It comprises ganglionic plexuses (GP), usually located in the epicardial fat layer [112]. The GP can be identified using high-frequency stimulation or anatomical mapping methods. PVI combined with endocardial GP ablation may enhance the efficacy of persistent AF ablation: randomized controlled trial enrolling 264 patients with persistent AF showed that the PVI combined with GP ablation group had a better rate of sinus rhythm maintenance than the PVI combined with linear ablation group at 3-year follow-up (49% vs 34%; *p* = 0.035) [113]. This finding has not been consistently replicated due to variations in study populations and methodologies [114,115]. Thermal ablation energy during PVI is often accompanied by peri-atrial GP ablation, impacting AF ablation success. Conversely, PFA, with its high selectivity, minimally affects GP during PVI [116]. Recent clinical trials show that the 1-year efficacy of PFA for treating persistent AF is comparable to or better than conventional thermal ablation [103,117]. The potential of PFA in the field of adjunctive GP ablation warrants further investigation.

### 4.3. Map-Guided Approach

(1).Low-voltage area (LVA) ablation

Atrial fibrosis is an important substrate for persistent AF reentry activity [118,119], which plays a significant role in the onset and progression of AF. These changes are specifically characterized by alterations in voltage and local potentials. Voltage mapping-guided ablation prevents overablation and aids operators in targeting arrhythmogenic areas. Previous clinical studies have demonstrated that ablation of low voltage areas could improve persistent AF clinical outcomes [120,121]. However, the recent conclusions from the DECAAF II trial in 2022 were disappointing: MRI-guided fibrotic areas ablation did not have a significant effect on the rate of recurrence of persistent atrial fibrillation [122]. But as cardiac MRI techniques use image intensity thresholds and do not normalise the signal. This makes it lack objectivity and reproducibility in identifying fibrotic areas [123]. Likewise, the STABLE-SR-II Trial randomized 300 patients with persistent AF into a CPVI-only group (*n* = 150) and a group undergoing ablation of CPVI + LVAs during sinus rhythm (SR) (*n* = 150). Follow-up results confirm that targeting LVA offers no additional benefit over CPVI alone in ’early’ PeAF patients [124]. RCTs on voltage-guided ablation are summarised in Table 2. Conflicting data from current studies on low-voltage area ablation have been investigated. The investigators speculate due to the diffuse nature of atrial myocardial lesions, ablation targeting low-voltage areas may not be able to alter the process. A more comprehensive substrate-modification approach may be necessary in the future. On the other hand, the definition of fibrotic ablation lacks standardization and established endpoints for fibrosis ablation procedures are absent, which makes this targeting strategy somewhat subjective.

(2).Rotor mapping and ablation

A rotor is a vortex generated by spin motion in a two-dimensional plane that generates spiral waves [130]. It is a key driver of AF. Ablation targeting the rotors has shown improved clinical outcomes in patients with persistent AF. Relevant clinical studies are shown in Table 3. Narayan introduced a new method with a 64-pole basket electrode and an innovative mapping technique to pinpoint the rotor region for guiding ablation. Following a median of 273 days of post-ablation, the FIRM group had a considerably greater success rate than the control group (82.4% vs. 44.9%; *p* < 0.001) [131]. Long-term follow-up success rates were also significantly higher in the FIRM (Focal Impulse and Rotor Modulation) group [132]. However, these results have not been replicated in later studies [133,134,135]. Intracardiac panoramic mapping has drawbacks. Although the 64-pole basket electrodes can provide comprehensive atrial mapping, the basket catheter fails to adequately cover the atrium, leaving up to half of the area unsampled and resulting in poor local resolution [136]. Follow-up studies indicate that the electrode spacing in high-density catheters is below the minimum resolution, enhancing their ability to localize rotors. However, this method has limitations, as it lacks panoramic mapping and cannot show overall atrial electrical activity.

The 2020 introduction of the RADAR system renewed optimism for rotor mapping [141]. The RADAR system needs to be used in conjunction with the conventional mapping system. Electrograms of different anatomical sites are recorded. Electrograms and their spatial data are processed by the RADAR system to generate a 3D panoramic vector map that identifies the AF driving domains. A total of 82% of patients who successfully terminated atrial fibrillation during ablation remained free of recurrence during follow-up.

Haissaguerre et al. utilized non-invasive bedside mapping methods for guiding ablation [142]. This method involved the patient wearing a vest embedded with 252 electrodes to monitor ECG activity continuously. Analysis of the patient’s surface signals helped in identifying rotor locations for guiding ablation. Atrial fibrillation-free recurrence rate is up to 85% after one year of ablation. Notwithstanding, the non-invasive system evaluates rotor sites based on distant unipolar signals which might adversely affect signal quality and resolution.

(3).Non-pulmonary vein triggers

Extrapulmonary venous trigger foci, which account for 10–20% of patients, are ectopic beats outside the pulmonary veins triggering atrial fibrillation post-PVI [143,144,145]. These foci are commonly located in specific areas like the posterior left atrial wall, interatrial septum, left auricle, and others. Ablation of non-pulmonary vein triggers improves prognosis in patients with persistent atrial fibrillation [146], as recommended by current expert consensus [147]. Stimulation manipulations aimed at inducing these triggers involve initiating high-dose isoprenaline (starting at 3 mg and incrementally increasing to 6 mg, 12 mg, and 20–30 mg every 3–5 min based on the heart rate response). Subsequently, a multipolar-catheters mapping of non-pulmonary vein trigger points is conducted [148]. Identifying these triggers is challenging due to their transient nature. Recently, Thind et al. developed an algorithm based on P-wave morphology using intracardiac activation sequences from linear decapolar catheters to achieve accurate regionalization of NPVT [149], which facilitates targeted mapping. However, P-wave morphology may not be reliable in patients with persistent AF with significant atrial remodeling in the later stages of the disease or post-catheter ablation. Recent studies have shown minimal differences in P-wave morphology following PVI compared to PWI, demonstrating its value in identifying non-pulmonary vein trigger foci. The P-wave algorithm created from the findings predicted the site of non-pulmonary vein triggers with an accuracy of up to 93% [150].

### 4.4. Surgical Epicardial Ablation and Hybrid Ablation

PVI combined with adjunctive ablation measures yielded mixed results in patients with persistent versus those with long-standing persistent atrial fibrillation. Achieving durable transmural lesions with conventional endocardial catheter ablation is challenging.

(1).Cox maze procedure

The Cox maze (CM) procedure, developed by James Cox and first applied clinically in 1987. It makes incisions in the left and right atria to form scar lines that interrupt macro-reentry circuits, thereby eliminating AF [151]. Due to the high rate of postoperative complications and the difficulty of the procedure, the original Cox-Maze procedure has evolved into the Cox-Maze III (CM-III) procedure after two refinements. The operative mortality rate for 118 patients who underwent maze III surgery was 2%, with 93% remaining free of atrial fibrillation after 8.5 years of follow-up [152]. In 2002, Melby and colleagues introduced the CM IV procedure, which uses bipolar radiofrequency combined with cryoablation as an alternative to replace the ‘cut and sew’ damage of the CM-III procedure [153]. Clinical results show that CM IV achieves the high success rates of CM III while significantly decreasing operative time and complication rates.

(2).Hybrid Approaches

The effectiveness of the hybrid procedure (closed-chest epicardial ablation combined with endocardial ablation) for treating persistent atrial fibrillation is garnering attention. Epicardial ablation targeted the posterior wall of the left atrium and the pulmonary veins, followed by endocardial ablation to complete the PVI and address any remaining gaps. As epicardial ablation is performed sequentially with endocardial ablation, conduction gaps that are not identified during endocardial ablation can be managed accordingly to achieve bidirectional conduction blockade. The hybrid approach enhances sinus rhythm maintenance in persistent AF patients without increasing adverse events compared to catheter ablation alone [154,155].

## 5. Conclusions

Despite extensive clinical studies, there is no clear consensus on the optimal adjunctive ablation strategy for patients with persistent atrial fibrillation. Recurrence of atrial fibrillation after ablation may be due to incomplete pulmonary vein isolation or other mechanisms beyond pulmonary vein triggering. Hence, more advanced ablation tools are required to isolate pulmonary veins with durable transmural ablation lesions. Furthermore, due to the presence of atrial remodeling and substrate formation in persistent atrial fibrillation, antiarrhythmic drugs or catheter ablation therapy alone are not enough. Current treatment strategies for persistent AF should be individualised and holistic. Multidisciplinary treatment options: intensive risk factor management, stroke prevention, and rhythm control combined with heart rate control can further improve the prognosis of patients with persistent AF.

## Figures and Tables

**Figure 1 biomedicines-13-00091-f001:**
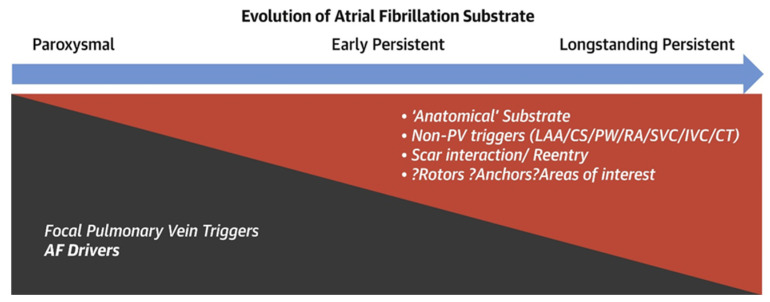
The difference in mechanism between paroxysmal and persistent atrial fibrillation; left atrial appendage LAA indicates left atrial appendage, CS indicates coronary sinus, PW indicates posterior wall; SVC indicates superior vena cava, IVC indicates inferior vena cava, RA indicates right atrium, CT indicates crista terminalis [4].

**Figure 2 biomedicines-13-00091-f002:**
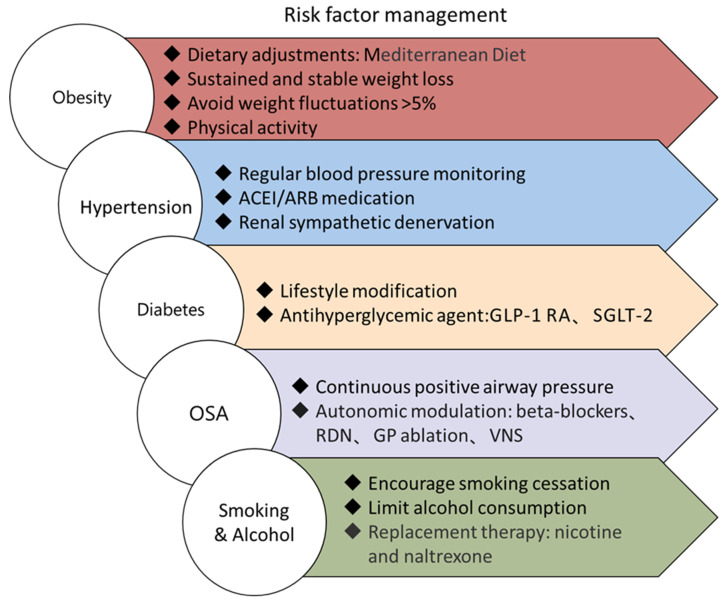
Risk factors for atrial fibrillation. GLP-1RAs: Glucagon-like peptide-1 receptor agonists; SGLT-2: Sodium-Glucose Transport Protein-2; OSA: Obstructive sleep apnea; RDN: Renal denervation; GP: Ganglionated plexi; VNS: Vagal nerve stimulation.

**Table 1 biomedicines-13-00091-t001:** Randomized study of PVI + PWI * in persistent AF.

Study (Year)	Participants	PeAF (%)	Randomization	Sinus Rhythm Outcome (%)		Follow Up Time
				Intervention	Control	
2023 (The CAPLA study) [99]	338	100	PVI + PWI vs. PVIalone	52.4	53.6	12 m
2022 [104]	100	100	PVI + PWI vs. PVIalone	76	54	457.9 ± 61.8 d
2021 [105]	110	100	PVI + PWI vs. PVIalone	74.5	54.5	12 m
2019 (POBI-AF) [100]	217	100	PVI + posterior wall Box Isolation vs. PVIalone	73.5	76.2	16.2 ± 8.8 m
2015 [98]	120	100	PVI + PWI vs. PVIalone	83.3	63.3	12 m
2012 [106]	220	39	PVI + single-ring isolation vs. wide antral pulmonary vein isolation	67	64	2 y
2009 [107]	120	40	PVI + linear lesions along the LA roof vs. PVI + Left atrial posterior wall isolation	55	55	10 ± 4 m

* PWI: posterior wall isolation.

**Table 2 biomedicines-13-00091-t002:** Randomized study of voltage-guided ablation in persistent atrial fibrillation.

Study (Year)	Participant	PeAF (%)	Randomization	Sinus Rhythm Outcome (%)		Follow Up Time
				Intervention	Control	
2023 (CAPLA Substudy) [125]	210	100	PVI plus PWI with posterior wall LVAs vs. PVI alone	44.8	41.9	12 m
2022 (STABLE-SR-II) [124]	300	100	CPVI plus low-voltage area modification vs. CPVI alone	67.2	67.4	18 m
2022 (DECAAF II) [122]	815	100	PVI plus MRI-guided atrial fibrosis ablation vs. PVI alone	57	53.9	12–18 m
2022 (ERASE-AF) [126]	324	100	PVI vs. PVI plus individualized substrate ablation of atrial low-voltage myocardium	65	50	12 m
2022 [127]	152	100	PVI alone or PVI + PW ablation vs. voltage-guided ablation	64	34	60 m
2018 [121]	124	49	Low-voltage guided ablation + CPVI vs. PVI with (persistent AF) or without (paroxysmal AF) additional linear ablation	68	42	12 ± 3 m
STABLE-SR (2017) [128]	229	100	Low-voltage guided ablation + CPVI vs. CPVI + additional linear lesions and defragmentation	74	71.5	18 m
2014 [129]	124	100 (L-PeAF *)	CPVI + individualized substrate modification vs. Stepwise ablation	65.5	45	12 m

* L-PeAF: long-lasting persistent atrial fibrillation.

**Table 3 biomedicines-13-00091-t003:** Randomized study of Focal Impulse and Rotor Modulation guided ablation in persistent atrial fibrillation.

Study (Year)	Participants	PeAF (%)	Intervention	Sinus Rhythm Outcome (%) Time		Follow Up Time
				Intervention	Control	
2017 The Indiana University FIRM * Registry [137]	170	31	FIRMguided ablation	70	/	1 y (179 to 570 d)
2017(AFACART study) [138]	118	100	Non-invasive mapping guided ablation targeted drivers + PVI	76.8	/	12 m
2016 [139]	68	100	Nonlinear phase mapping- guided substrate ablation vs. extensive complex fractionated atrial electrograms ablation	58.3 (group-1)	77.3 (group-2)	17.7 ± 8.17 m
2015 [133]	29	100	FIRM-guided only ablation	17	/	5.7 m
2013 [140]	73	49.3	Conventional ablation vs. FIRM plus conventional ablation	74	39	890 d
CONFIRM (2012) [131]	92	70.7	FIRM + conventional ablation vs. conventional ablation alone	82.4	44.9	273 d (132–681)

* FIRM: Focal Impulse and Rotor Modulation.

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
