# Peer review of "Integrated Management of Persistent Atrial Fibrillation"

_biomedicines, 2025, doi:10.3390/biomedicines13010091_

Round 1

Reviewer 1 Report

Comments and Suggestions for Authors

Article « Integrated management of persistent atrial fibrillation». Authors: Xindi Yue etc. In this review provides a comprehensive review of the management strategies for persistent atrial fibrillation (PeAF), an increasingly prevalent condition associated with significant morbidity. The authors discuss risk factor management, heart rate control, rhythm control, and emerging strategies such as adjunctive ablation techniques. The review addresses the challenges of PeAF associated with its treatment, particularly in the context of catheter ablation where success rates remain suboptimal.  The discussion on risk factors such as obesity, hypertension, and obstructive sleep apnea (OSA) is detailed and well-supported by recent evidence. The article is well structured, written in a clear and understandable language, the conclusions are logical, the literature corresponds to the stated topic. I recommend publishing.

Author Response

Thank you for your support. Your encouragement means a lot to us!

Reviewer 2 Report

Comments and Suggestions for Authors

This manuscript provides a comprehensive overview of current management strategies for persistent atrial fibrillation. It compasses risk factor modification, rate control and rhythm control. While the review is well-structured and covers a comprehensive analysis of AF management, it could benefit from a more in-depth exploration of certain key areas.

The authors appropriately emphasize lifestyle and risk factor modification, reflecting its class I recommendation in the ESC guidelines (Van Gelder European Heart Journal, Volume 45, Issue 36, 21 September 2024, Pages 3314–3414, https://doi.org/10.1093/eurheartj/ehae176). However, it is important to acknowledge that the evidence supporting risk factor management stems from moderate quality of evidence from non-randomized, observation trials and meta-analysis, particularly for smoking and hypertension.

The section on the rate control effectively highlights its important in overall AF management. There appears to be a syntax error on line 184 that needs correction (a full stop should be replaced by a comma).

For rhythm control, it should be highlighted that EAST-AFNET 4 trial primarily enrolled patients with paroxysmal atrial fibrillation. In the rhythm control arm, most patients received antiarrhythmic medications (87%) rather than ablation (19.6%). To my knowledge, no early rhythm control trials have specifically focused on the persistent AF population. Manuscript does not refer to use of anti-arrhythmic medications in restoring and maintenance of sinus rhythm. If weak evidence only then this should be cited as appropriate.

While the paper analyses several key trials in the empiric approach section, some areas lack critical evaluation. For instance, in the CAPLA trial, the lack of confirmation of posterior wall isolation using box isolation approach, with nearly 3/5 of patients failing to achieve durable posterior wall isolation raises concern. Protocol may not be strictly adhered in isolating posterior wall given risk of oesophageal injury (line 213). The cited meta-analysis (ref 91) indicates that patients with persistent AF benefit from adjunctive posterior wall isolation with a reduced risk of AF recurrence. It is unclear whether the lower recurrence rate in the cryoablation was related to posterior wall reconnection (line 223). Ablation energy may have a role to play in isolating the posterior wall effectively. In the table, there is a mixture of positive and negative trials. Posterior wall isolation may have a role in selective patient as per subgroup analysis.

Vein of Marshal ethanol ablation is an effective strategy in persistent AF ablation. However, coronary sinus reconnection is usually responsible for recurrence therefore combination of RF and ethanol ablation may be necessary to reduce risk of reconnection (line 242).

Additional points to consider – reference 102 should be checked for accuracy. Data do not match referred article (Line 254). The pulsed field ablation trial assessing ganglionic plexus had a follow up period of 3 months. Please cite PFA ablation study with >1 year follow up following GP ablation (Ref 105)(Line 259). The review should acknowledge the lack of reproducibility in atrial fibrosis using CMR imaging (line 270). Table 2 should include ERASE-AF  and study by Cutler et al (JACC 2022;33:2475-2484) findings.

It may be worth raising the possibility that surgical epicardial and hybrid ablation might be an option in selected group of patients who have failed to maintain sinus rhythm despite endocardial linear/non-PV trigger ablation. Potentially include line or two in an upcoming trials +/- ablation strategies to enhance the manuscript’s value.

By addressing these points, the authors can further improve the clarity, accuracy and impact of this article.

Author Response

   Thank you very much for your valuable suggestions, which improved our manuscript. We responded to each comment and made corresponding changes in the manuscript. According to these comments and suggestions, we added new information to the revised manuscript to help clarify the content. The revisions in the revised manuscriptare shown in red.

Comments 1: [However, it is important to acknowledge that the evidence supporting risk factor management stems from moderate quality of evidence from non-randomized, observation trials and meta-analysis, particularly for smoking and hypertension.]

Response 1: Thank you for pointing this out. I agree with this comment. Therefore, in the Risk Factors section, I have added some of the literature to support the evidence. Line 91-93, Line 96-98, Line 96-98, Line 100-102, Line 107-113,

Comments 2: [There appears to be a syntax error on line 184 that needs correction (a full stop should be replaced by a comma).]

Response 2: We feel sorry for our carelessness. In our resubmitted manuscript, the syntax error is revised. Thanks for yourcorrection.  line 197

Comments 3: [For rhythm control, it should be highlighted that EAST-AFNET 4 trial primarily enrolled patients with paroxysmal atrial fibrillation. In the rhythm control arm, most patients received antiarrhythmic medications (87%) rather than ablation (19.6%).]

Response 3: [Thank you for your constructive feedback. Specifically, we have added a more in-depth analysis of the results of the EAST-AFNET 4 trial.  line 213-219

Comments 4: [Manuscript does not refer to use of anti-arrhythmic medications in restoring and maintenance of sinus rhythm. If weak evidence only then this should be cited as appropriate.]

Response 4: Thank you for your suggestion. We have added a discussion on the current status of antiarrhythmic drugs in atrial fibrillation therapy. line 221-231

Comments 5: [While the paper analyses several key trials in the empiric approach section, some areas lack critical evaluation. For instance, in the CAPLA trial, the lack of confirmation of posterior wall isolation using box isolation approach, with nearly 3/5 of patients failing to achieve durable posterior wall isolation raises concern. Protocol may not be strictly adhered in isolating posterior wall given risk of oesophageal injury (line 213).]

Response 5: Your suggestions are very constructive. We have added an in-depth discussion and critical analysis of the findings of the study (especially the CAPLA trial) (lines 253 to 260, lines 266 to 271). With these changes, we hope to present a fuller picture of the significance and contribution of the study.

Comments 6: [Vein of Marshal ethanol ablation is an effective strategy in persistent AF ablation. However, coronary sinus reconnection is usually responsible for recurrence therefore combination of RF and ethanol ablation may be necessary to reduce risk of reconnection]

Response 6: We agree with this comment and refine the relevant content in the revised manuscript. lines 280 to 284

Comments 7: [Additional points to consider – reference 102 should be checked for accuracy. Data do not match referred article (Line 254).]

Response 7: We were really sorry for our careless mistakes. Thank you for your reminder. (Line 306-307)

Comments 8: [The pulsed field ablation trial assessing ganglionic plexus had a follow up period of 3 months. Please cite PFA ablation study with >1 year follow up following GP ablation (Ref 105) (Line 259).]

Response 8: The authors apologise for confusing the reviewers' understanding of Original Research: Pulsed Field Ablation to Treat Atrial Fibrillation: Autonomic Nervous System Effects. In order to make it clearer to the reviewers, the corresponding context has been reorganised in the revised manuscript. Pulsed field ablation still achieves a success rate comparable to thermal ablation with less impact on autonomic nervous system. The potential of PFA in the field of adjunctive GP ablation warrants further investigation. (Line 310-313)

Comments 9: [The review should acknowledge the lack of reproducibility in atrial fibrosis using CMR imaging (line 270). Table 2 should include ERASE-AF and study by Cutler et al (JACC 2022;33:2475-2484) findings.]

Response 9: We appreciate your thorough review of ourmanuscript. We add a critical appraisal of the field of LVA by discussing the limitations of CMR in detecting atrial fibrosis. (Line 324-326). In response to your suggestion, we have updated the references section in the Table 2.

Comments 10: [It may be worth raising the possibility that surgical epicardial and hybrid ablation might be an option in selected group of patients who have failed to maintain sinus rhythm despite endocardial linear/non-PV trigger ablation. Potentially include line or two in an upcoming trials +/- ablation strategies to enhance the manuscript’s value.]

Response 10: As suggested by the reviewers, we have added to the section on Surgical epicardial ablation and hybrid ablation. (Line 402-423)

  Thanks to the professional comments again that point out the above problems. The authors hope these explanations would answer your doubts.

Reviewer 3 Report

Comments and Suggestions for Authors

The article submitted for review is a review article on the integrated management of persistent atrial fibrillation. The topic targeted for this review is very specific but of great interest. Indeed, atrial fibrillation increases the risk of heart failure, stroke, and mortality. This review provides a current state of the art and the latest treatments for persistent atrial fibrillation in the areas of rhythm control, heart rate control and risk factor management.
This review is perfectly constructed both in methodology and data presentation. It may be necessary to improve the figures, in order to make them more scientific or to consider them as graphical abstract.
The discussion of the various references is perfectly done and this with some critical aspects. The conclusion must also be completed by more developed perspectives.

Author Response

  We sincerely appreciate your valuable comments! We use this feedback to improve the quality of manuscripts. The paper has been revised carefully and thoroughly according to the comments. Changes/additions to the manuscript are highlighted in red text.

Comments 1: [It may be necessary to improve the figures, in order to make them more scientific or to consider them as graphical abstract.]

Response 1: Thank you very much for your constructive comments. We have improved the article picture. We hope it outlines the full review and more scientific. (Graphical abstract).

Comments 2: [The conclusion must also be completed by more developed perspectives.]

Response 2: We think this is an excellent suggestion. We have re-written the conclusion section according to your suggestion to better summarize the review(Line430-434).

   Thanks to the professional comments again that point out the above problems. The authors hope these explanations would answer your doubts.